# VARIATIONAL BI-LSTMS

## ABSTRACT

Recurrent neural networks like long short-term memory (LSTM) are important architectures for sequential prediction tasks. LSTMs (and RNNs in general) model sequences along the forward time direction. Bidirectional LSTMs (Bi-LSTMs), which model sequences along both forward and backward directions, generally perform better at such tasks because they capture a richer representation of the data. In the training of Bi-LSTMs, the forward and backward paths are learned independently. We propose a variant of the Bi-LSTM architecture, which we call Variational Bi-LSTM, that creates a dependence between the two paths (during training, but which may be omitted during inference). Our model acts as a regularizer and encourages the two networks to inform each other in making their respective predictions using distinct information. We perform ablation studies to better understand the different components of our model and evaluate the method on various benchmarks, showing state-of-the-art performance.

## 1 INTRODUCTION

Recurrent neural networks (RNNs) have become the standard models for sequential prediction tasks, having achieved state of the art performance in a number of applications that includes sequence prediction, language translation, machine comprehension, and speech synthesis (Arik et al., 2017; Wang et al., 2017; Mehri et al., 2016; Sotelo et al., 2017). RNNs model temporal data by encoding a given arbitrary-length input sequentially, at each time step combining some transformation of the current input with the encoding from the previous time step. This encoding, referred to as the RNN hidden state, summarizes all previous input tokens.

Viewed as "unrolled" feedforward networks, RNNs can become arbitrarily deep depending on the input sequence length, and use a repeating module to combine the input with the previous state at each time step. Consequently, they suffer from the vanishing/exploding gradient problem (Pascanu et al., 2012). This problem has been addressed through architectural variants like the long short-term memory (LSTM) (Hochreiter & Schmidhuber, 1997) and the gated recurrent unit (GRU) (Chung et al., 2014). These architectures add a linear path along the temporal sequence which allows gradients to flow more smoothly back through time.

Various regularization techniques have also been explored to improve RNN performance and generalization. Dropout (Srivastava et al., 2014) regularizes a network by randomly dropping hidden units during training. However, it has been observed that using dropout directly on RNNs is not as effective as in the case of feed-forward networks. To combat this, Zaremba et al. (2014) propose to instead apply dropout on the activations that are not involved in the recurrent connections (Eg. in a multi-layer RNN); Gal & Ghahramani (2016) propose to apply the same dropout mask through an input sequence during training. In a similar spirit to dropout, Zoneout (Krueger et al., 2016) proposes to choose randomly whether to use the previous RNN hidden state.

The aforementioned architectures model sequences along the forward direction of the input sequence. Bidirectional-LSTM, on the other hand, is a variant of LSTM that simultaneously models each sequence in both the forward and backward directions. This enables a richer representation of data, since each token's encoding contains context information from the past and the future. It has been shown empirically that bidirectional architectures generally outperform unidirectional ones on many sequence-prediction tasks. However, the forward and backward paths in Bi-LSTMs are trained separately and the benefit usually comes from the combined hidden representation from both paths. In this paper, our main idea is to create a dependence between the two paths that acts as a regulariza-

tion during training, but doesn't hinder inference even in the absence of the other path (the backward path in practical scenarios). We note that recently proposed methods like TwinNet Serdyuk et al. (2017) and Z-forcing Sordoni et al. (2017) are similar in spirit to this idea. In our approach, we use a variational auto-encoder (VAE; Kingma & Welling (2014)) that takes as input the hidden states from the two paths of the Bi-LSTM and maps them to a shared hidden representation of the VAE at each time step. The samples from the VAE hidden state are then used both for reconstructing the LSTM hidden states and feeding forward to the next hidden state. In this way, we create a *channel* between the two paths that acts as a regularization for learning better representations. We refer to the resulting model as a Variational Bi-LSTM.

Below, we describe Variational Bi-LSTMs in detail and then demonstrate empirically their ability to model complex sequential distributions. In experiments, we obtain state-of-the-art or competitive performance on the tasks of Penn Treebank, IMDB, TIMIT, Blizzard, and Sequential MNIST.

## 2 VARIATIONAL BI-LSTM

Bi-LSTM is a powerful architecture for sequential tasks because it models temporal data both in the forward and backward directions. For this it uses two LSTMs that are generally learned independently of each other; the richer representation results from combining the hidden states of these LSTMs, where combination is often by concatenation. The idea behind variational Bi-LSTMs is to create a *channel* of information exchange between the two LSTMs that helps the model to learn better representations. We create this dependence using a variational auto-encoder (VAE). This enables us to take advantage of the fact that VAE allows for sampling from a prior during inference. For sequence prediction tasks like language generation, while one can use Bi-LSTMs during training, there is no straightforward way to employ the full bidirectional model during inference – this would involve, eg, generating a sentence starting at both its beginning and end. In such cases, the VAE allows us to sample from the prior at inference time to make up for the absence of the backward LSTM.

Now we describe our variational Bi-LSTM model formally. Let $\mathbf{X} = \{\mathbf{x}^{(i)}\}_{i=1}^N$ be a dataset consisting of $N$ i.i.d. sequential data samples of continuous or discrete variables. For notational convenience, we will henceforth drop the superscript $i$ indexing samples. For each sample sequence $\mathbf{x} = (\mathbf{x}_1, \ldots, \mathbf{x}_T)$, the hidden state of the forward LSTM is given by:

$$\mathbf{h}_t = \overrightarrow{f}(\mathbf{x}_t, \mathbf{h}_{t-1}, \mathbf{z}_t, \tilde{\mathbf{b}}_t).$$

The hidden state of the backward LSTM is given by,

$$\mathbf{b}_t = \overleftarrow{f}(\mathbf{x}_t, \mathbf{b}_{t+1}).$$

In both cases, the function $f$ represents the standard LSTM updates, modified to account for the additional arguments.

In the forward LSTM model, we introduce three latent random variables, $\mathbf{z}_t$, $\tilde{\mathbf{b}}_t$, and $\tilde{\mathbf{h}}_{t-1}$, where $\mathbf{z}_t$ depends on $\mathbf{h}_{t-1}$ and $\mathbf{b}_t$ during training, and $\tilde{\mathbf{b}}_t$ and $\tilde{\mathbf{h}}_{t-1}$ depend only on $\mathbf{z}_t$ (see figure 1-left, for a graphical representation). Note that so far, $\tilde{\mathbf{b}}_t$ and $\tilde{\mathbf{h}}_{t-1}$ are simply latent vectors drawn from conditional distributions $p_\psi$ and $p_\xi$, respectively, to be defined below. However, as explained in Section 2.1 (see also dashed lines in figure 1-left), we will encourage these to lie near the manifolds of backward and forward LSTM states, respectively.

By design, the joint conditional distribution over latent variables $\mathbf{z}_t$ and $\tilde{\mathbf{b}}_t$ with parameters $\theta$ and $\psi$ factorizes as $p_\theta(\mathbf{z}_t | \mathbf{x}_{1:t-1}, \mathbf{z}_{1:t-1}) p_\psi(\tilde{\mathbf{b}}_t | \mathbf{z}_t)$. This factorization enables us to formulate several helpful auxiliary cots, as defined in the next subsection. Further, $p_\eta(\mathbf{x}_{t+1} | \mathbf{x}_{1:t}, \mathbf{z}_{1:t}, \tilde{\mathbf{b}}_t)$ defines the generating model, which induces the distribution over the next observation given the previous states and the current input.

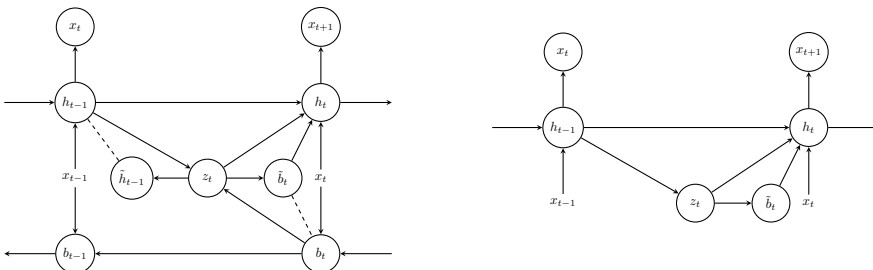

(a) Training phase of variational Bi-LSTM      (b) Inference phase of variational Bi-LSTM

Figure 1: Graphical description of our proposed variational Bi-LSTM model during train phase (left) and inference phase (right). During training, each step $t$ is composed of an encoder which receives both the past and future summary via $\mathbf{h}_{t-1}$ and $\mathbf{b}_t$ respectively, and a decoder that generates $\tilde{\mathbf{h}}_{t-1}$ and $\tilde{\mathbf{b}}_t$ which are forced to be close enough to $\mathbf{h}_{t-1}$ and $\mathbf{b}_t$ using two auxiliary reconstruction costs (dashed lines). This dependence between backward and forward LSTM through the latent random variable encourages the forward LSTM to learn a richer representation. During inference, the backward LSTM is removed. In this case, $\mathbf{z}_t$ is sampled from the prior as in a typical VAE, which in our case, is defined as a function of $\mathbf{h}_{t-1}$.

Then the marginal likelihood of each individual sequential data sample $\mathbf{x}$ can be written as

$$
\begin{aligned}
p(\mathbf{x}; \mathbf{\Gamma}) = \prod_{t=0}^{T} p(\mathbf{x}_{t+1}|\mathbf{x}_{1:t}) &= \prod_{t=0}^{T} \int_{\mathbf{z}_{1:T}} p(\mathbf{x}_{t+1}|\mathbf{x}_{1:t}, \mathbf{z}_{1:t}) p_\theta(\mathbf{z}_t|\mathbf{x}_{1:t-1}, \mathbf{z}_{1:t-1}) d\mathbf{z}_{1:T} \\
&= \prod_{t=0}^{T} \int_{\mathbf{z}_{1:T}} \int_{\tilde{\mathbf{b}}_t} \Big[ p_\eta(\mathbf{x}_{t+1}|\mathbf{x}_{1:t}, \mathbf{z}_{1:t}, \tilde{\mathbf{b}}_t) p_\psi(\tilde{\mathbf{b}}_t|\mathbf{z}_{1:t}) p_\theta(\mathbf{z}_t|\mathbf{x}_{1:t-1}, \mathbf{z}_{1:t-1}) \Big] d\tilde{\mathbf{b}}_t d\mathbf{z}_{1:T},
\end{aligned}
\tag{1}
$$

where $q_\phi(\mathbf{z}_t|\mathbf{x})$ is the conditional inference model and $\mathbf{\Gamma} = \{\phi, \theta, \psi, \eta\}$ is the set of all parameters of the model. Here, we assume that all conditional distributions belong to parametrized families of distributions which can be evaluated and sampled from efficiently.

Note that the joint distribution in equation (1) is intractable. Kingma & Welling (2014) demonstrated how to maximize a variational lower bound, $\mathcal{L}_\mathbf{\Gamma}$, of the data log likelihood instead, which is given by

$$
\begin{aligned}
\log p(\mathbf{x}; \mathbf{\Gamma}) \geq \mathcal{L}_\mathbf{\Gamma} = \sum_{t=0}^{T} \underset{\mathbf{z}_{1:T} \sim q_\phi(\mathbf{z}|\mathbf{x})}{E} \underset{\tilde{\mathbf{b}}_t \sim p_\psi(\tilde{\mathbf{b}}_t|\mathbf{z}_t)}{E} \Big[ \log p_\eta(\mathbf{x}_{t+1}|\mathbf{x}_{1:t}, \mathbf{z}_{1:t}, \tilde{\mathbf{b}}_t) \Big] \\
- D_{KL}(q_\phi(\mathbf{z}_t|\mathbf{x}) \| p_\theta(\mathbf{z}_t|\mathbf{x}_{1:t-1}, \mathbf{z}_{1:t-1})),
\end{aligned}
\tag{2}
$$

where $D_{KL}$ is the Kullback-Leibler (KL) divergence between the approximate posterior and the conditional prior (see the appendix). This is the approach we take.

## 2.1 TRAINING AND INFERENCE

In the proposed variational Bi-LSTM, the latent variable $\mathbf{z}_t$ is inferred as

$$
\mathbf{z}_t \sim q_\phi(\mathbf{z}_t|(\mathbf{h}_{t-1}, \mathbf{b}_t)) = \mathcal{N}(\boldsymbol{\mu}_{q,t}, \mathrm{diag}(\boldsymbol{\sigma}_{q,t}^2)),
\tag{3}
$$

in which $[\boldsymbol{\mu}_{q,t}, \boldsymbol{\sigma}_{q,t}^2] = f_\phi(\mathbf{h}_{t-1}, \mathbf{b}_t)$ where $f_\phi$ is a multi-layered feed-forward network with Gaussian outputs. We assume that the prior over $\mathbf{z}_t$ is a diagonal multivariate Gaussian distribution given by

$$
p_\theta(\mathbf{z}_t|\mathbf{x}_{1:t-1}, \mathbf{z}_{1:t-1}) = \mathcal{N}(\boldsymbol{\mu}_{p,t}, \mathrm{diag}(\boldsymbol{\sigma}_{p,t}^2)), \quad \text{where} \quad [\boldsymbol{\mu}_{p,t}, \boldsymbol{\sigma}_{p,t}^2] = f_\theta(\mathbf{h}_{t-1}),
\tag{4}
$$

for a fully connected network $f_\theta$. This is important because, during generation (see Figure 1-right, for a graphical representation), we will not have access to the backward LSTM. In this case, as in a

VAE, we will sample from the prior for $\mathbf{z}_t$. Since we define the prior to be a function of $\mathbf{h}_{t-1}$, the forward LSTM is encouraged during training to learn the dependency due to the backward hidden state $\mathbf{b}_t$.

The latent variable $\tilde{\mathbf{b}}_t$ is meant to model information coming from the future of the sequence. Its conditional distribution is given by

$$p_\psi(\tilde{\mathbf{b}}_t|\mathbf{z}_t) = \mathcal{N}(\boldsymbol{\mu}_{\tilde{\mathbf{b}},t}, \text{diag}(\boldsymbol{\sigma}_{\tilde{b},t}^2)), \tag{5}$$

where $[\boldsymbol{\mu}_{\tilde{\mathbf{b}},t}, \boldsymbol{\sigma}_{\tilde{\mathbf{b}},t}^2] = f_\psi(\mathbf{z}_t)$ for a fully connected neural network $f_\psi$ (See Figure 1(a)). To encourage the encoding of future information in $\tilde{\mathbf{b}}_t$, we maximize the probability of the true backward hidden state, $\mathbf{b}_t$, under the distribution $p_\psi$, as an auxiliary cost during training. In this way we treat $\tilde{\mathbf{b}}_t$ as a predictor of $\mathbf{b}_t$, similarly to what was done by Sordoni et al. (2017).

To capture information from the past in the latents, we similarly use $\tilde{\mathbf{h}}_{t-1}$ as a predictor of $\mathbf{h}_{t-1}$. This is accomplished by maximizing the probability of the latter under the conditional distribution of the former, $\log p_\xi(\tilde{\mathbf{h}}_{t-1}|\mathbf{z}_t)$, as another auxiliary cost, where

$$p_\xi(\tilde{\mathbf{h}}_{t-1}|\mathbf{z}_t) = \mathcal{N}(\boldsymbol{\mu}_{\tilde{\mathbf{h}},t}, \text{diag}(\boldsymbol{\sigma}_{\tilde{\mathbf{h}},t}^2)). \tag{6}$$

Here, $[\boldsymbol{\mu}_{\tilde{\mathbf{h}},t}, \boldsymbol{\sigma}_{\tilde{\mathbf{h}},t}^2]$ is the output of a fully-connected neural network $f_\xi$ taking $\mathbf{z}_t$ as input. The auxiliary costs arising from distributions $p_\xi$ and $p_\psi$ teach the variational Bi-LSTM to encode past and future information into the latent space of $\mathbf{z}$.

We assume that parameters of the generating distribution $p_\eta(\mathbf{x}_{t+1}|\mathbf{x}_{1:t}, \mathbf{z}_{1:t}, \tilde{\mathbf{b}}_t)$ are computed via MLP, taking the form of either a Gaussian distribution output in the continuous case or categorical proportions output in the discrete (ie, one-hot) prediction case (See Figure 1(b)).

All the parameters in $\boldsymbol{\Gamma}$ and $\boldsymbol{\xi}$ are updated based on backpropagation through time (Rumelhart et al., 1988) using the reparameterization trick (Kingma & Welling, 2014), where the gradients are computed by differentiating of the following function:

$$\mathcal{L}(\mathbf{x}; \boldsymbol{\Gamma}, \boldsymbol{\xi}) = \sum_{t=0}^{T} \mathop{E}_{z_{1:T} \sim q_\phi(\mathbf{z}|\mathbf{x})} \Big[ \mathop{E}_{\tilde{\mathbf{b}}_t \sim p_\psi(b|\mathbf{z}_{1:t})} \Big[ \log p(\mathbf{x}_{t+1}|\mathbf{x}_{1:t}, \mathbf{z}_{1:t}, \tilde{\mathbf{b}}_t) + $$
$$\alpha \log p_\psi(\mathbf{b}_t|\mathbf{z}_t) + \beta \log p_\xi(\mathbf{h}_{t-1}|\mathbf{z}_t) \Big]\Big] - $$
$$D_{KL}(q_\phi(\mathbf{z}_t|\mathbf{x})\|p_\theta(\mathbf{z}_t|\mathbf{x}_{1:t-1}, \mathbf{z}_{1:t-1})).$$

Here, $\alpha$ and $\beta$ are non-negative real numbers. We improve training convergence with a trick for the variational Bi-LSTM, which we refer to as skip gradient, meant to ease learning of the latent variables. It is well known that autoregressive decoder models tend to ignore their stochastic variables (Bowman et al., 2015). Skip gradient is a technique to encourage that relevant summaries of the past and the future are encoded in the latent space. The idea is to skip the gradient of the stochastic operations *with respect to the recurrent units* through time. To achieve this, at each time step, a mask drawn from a Bernoulli distribution governs whether to skip the gradient or to backpropagate it for a given data point.

## 3 EXPERIMENTAL RESULTS

In this section demonstrate the effectiveness of our proposed model on several tasks. We present experimental results obtained when training the Variational Bi-LSTM on various sequential datasets: Penn Treebank (PTB), IMDB, TIMIT, Blizzard, and Sequential MNIST. Our main goal is to ensure that the model proposed in Section 2 can benefit from a generated relevant summary of the future that yields competitive results. In all experiments, we train all the models using ADAM optimizer (Kingma & Ba, 2014) and we set all MLPs in Section 2 to have one hidden layer with leaky-ReLU hidden activation. All the models are implemented using Theano (Theano Development Team, 2016) and the code is available at `https://anonymous.url`.

**Blizzard:** Blizzard is a speech model dataset with 300 hours of English, spoken by a single female speaker. We report the average log-likelihood for half-second sequences (Fraccaro et al., 2016). In

our experimental setting, we use 1024 hidden units for MLPs, 1024 LSTM units and 512 latents. Our model is trained using learning rate of 0.001 and minibatches of size 32 and we set $\alpha = \beta = 1$. A fully factorized multivariate Gaussian distribution is used as the output distribution. The final lower bound estimation on TIMIT can be found in Table 1.

Table 1: The average of log-likelihood per sequence on Blizzard and TIMIT testset

| Model | Blizzard | TIMIT |
|---|---|---|
| RNN-Gauss | 3539 | -1900 |
| RNN-GMM | 7413 | 26643 |
| VRNN-I-Gauss | $\geq$ 8933 | $\geq$ 28340 |
| VRNN-Gauss | $\geq$ 9223 | $\geq$ 28805 |
| VRNN-GMM | $\geq$ 9392 | $\geq$ 28982 |
| SRNN (smooth+res$_q$) | $\geq$ 11991 | $\geq$ 60550 |
| Z-Forcing (Sordoni et al., 2017) | $\geq$ 14315 | $\geq$ 68852 |
| Variational Bi-LSTM | $\geq$ **17319** | $\geq$ **73315** |

**TIMIT:** Another speech modeling dataset is TIMIT with 6300 English sentences read by 630 speakers. Like the work done in (Fraccaro et al., 2016), our model is trained on raw sequences of 200 dimensional frames. In our experiments, we use 1024 hidden units, 2048 LSTM units and 128 latent variables, and batch size of 128. We train the model using learning rate of 0.0001, $\alpha = 1$ and $\beta = 0$. The average log-likelihood for the sequences on test can be found in Table 1.

**Sequential MNIST:** We use the MNIST dataset which is binarized according to (Murray & Salakhutdinov, 2009) and we downloaded in binrized from (Larochelle, 2011). Our best model consists of 1024 hidden units, 1024 LSTM units and 256 latent variables. We train the model using a learning rate of 0.0001 and a batch size of 32. To reach the negative log-likelihood reported in Table 2, we set $\alpha = 0.001$ and $\beta = 0$.

Table 2: The average of negative log-likelihood on sequential MNIST

| Models | Seq-MNIST |
|---|---|
| DBN 2hl (Germain et al., 2015) | $\approx$ 84.55 |
| NADE (Uria et al., 2016) | 88.33 |
| EoNADE-5 2hl (Raiko et al., 2014) | 84.68 |
| DLGM 8 (Salimans et al., 2014) | $\approx$ 85.51 |
| DARN 1hl (Gregor et al., 2015) | $\approx$ 84.13 |
| BiHM (Bornschein et al., 2015) | $\approx$ 84.23 |
| DRAW (Gregor et al., 2015) | $\leq$ 80.97 |
| PixelVAE (Gulrajani et al., 2016) | $\approx$ **79.02** |
| Prof. Forcing (Goyal et al., 2016) | 79.58 |
| PixelRNN$_{(1\text{-layer})}$ (Oord et al., 2016) | 80.75 |
| PixelRNN$_{(7\text{-layer})}$ (Oord et al., 2016) | 79.20 |
| Z-Forcing (Sordoni et al., 2017) | $\leq$ 80.09 |
| Variational Bi-LSTM | $\leq$ 79.78 |

**IMDB:** It is a dataset consists of 350000 movie reviews (Diao et al., 2014) in which each sentence has less than 16 words and the vocabulary size is fixed to 16000 words. In this experiment, we use 500 hidden units, 500 LSTM units and latent variables of size 64. The model is trained with a batch size of 32 and a learning rate of 0.001 and we set $\alpha = \beta = 1$. The word perplexity on valid and test dataset is shown in Table 3.

**PTB:** Penn Treebank (Marcus et al. (1993)) is a language model dataset consists of 1 million words. We train our model with 1024 LSTM units, 1024 hidden units, and the latent variables of size 128. We train the model using a standard Gaussian prior, a learning rate of 0.001 and batch size of 50 and

Table 3: Word perplexity on IMDB on valid and test sets

| Model | Valid | Test |
|---|---|---|
| Gated Word-Char | 70.60 | 70.87 |
| Z-Forcing (Sordoni et al., 2017) | 56.48 | 65.68 |
| Variational Bi-LSTM | **51.43** | **51.60** |

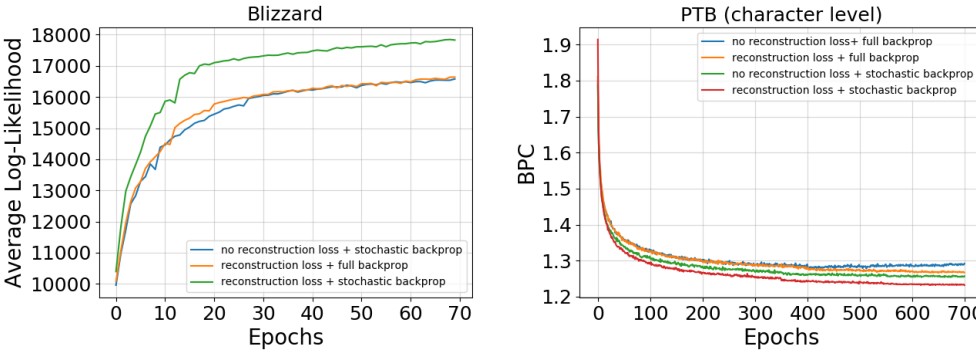

Figure 2: Evolution of the average of log-likelihood during training of Variational Bi-LSTMs with and without using skip gradient and auxiliary costs on PTB and Blizzard.

we set $\alpha = \beta = 1$. The model is trained to predict the next character in a sequence and the final bits per character on test and valid sets are shown in Table 4

Table 4: Bits Per Character (BPC) on PTB valid and test sets

| Model | Valid | Test |
|---|---|---|
| Unregularized LSTM | 1.47 | 1.36 |
| Weight noise | 1.51 | 1.34 |
| Norm stabilizer | 1.46 | 1.35 |
| Stochastic depth | 1.43 | 1.34 |
| Recurrent dropout | 1.40 | 1.29 |
| Zoneout (Krueger et al. (2016)) | 1.36 | 1.25 |
| RBN (Cooijmans et al. (2016)) | - | 1.32 |
| H-LSTM + LN (Ha et al. (2016)) | 1.28 | 1.25 |
| 3-HM-LSTM + LN (Chung et al., 2016) | - | 1.24 |
| 2-H-LSTM + LN (Ha et al. (2016)) | 1.25 | **1.22** |
| Z-Forcing | 1.29 | 1.26 |
| Variational Bi-LSTM | 1.26 | 1.23 |

## 4    ABLATION STUDIES

The goal of this section is to study the importance of the various components in our model to avoid any triviality. The experiments are as follows:

**1. Reconstruction loss on $h_t$ vs activity regularization on $h_t$**

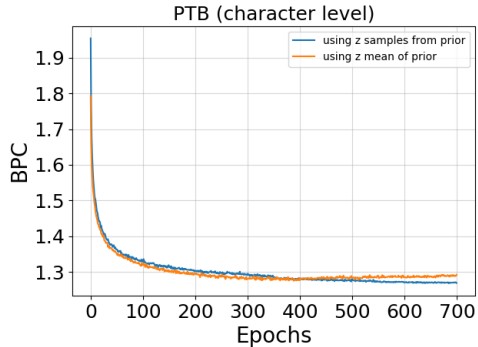

Figure 3: Evolution of the bits per character on PTB validation with sampling latent variables $\mathbf{z}$ from $\mathcal{N}(\mathbf{0}, \mathbf{I})$ during training or using a fixed vector which we set to be the mean of latent variables. Interestingly, not sampling from prior during inference does not hurt the final performance on PTB.

Table 5: Perplexity on IMDB using different coefficient $\gamma$ for activity regularization

| $\gamma$ | 0.001 | 1. | 4. | 8. | 16. |
|---|---|---|---|---|---|
| Test perplexity | 56.07 | 60.74 | 69.97 | 77.24 | 86.72 |

The authors of Merity et al. (2017) study the importance of activity regularization (AR) on the hidden states on LSTMs given as,

$$\mathcal{R}_{AR} = \gamma \|\mathbf{h}_t\|_2^2 \tag{7}$$

$$\tag{8}$$

However, since our model's reconstruction term on $\mathbf{h}_t$ can be decomposed as,

$$\|\mathbf{h}_t - \tilde{\mathbf{h}}_t\|_2^2 = \|\mathbf{h}_t\|_2^2 + \|\tilde{\mathbf{h}}_t\|_2^2 - 2\mathbf{h}_t^T \tilde{\mathbf{h}}_t \tag{9}$$

we perform experiments to confirm that the gains in our approach is not due to the $\ell^2$ regularization alone since our regularization encapsulates an $\ell^2$ term along with the dot product term.

We use activity regularization using hyperparameter $\alpha \in \{0.001, 1, 4, 8, 16\}$ in place of reconstruction term in our model and study the test perplexity. The results are shown in table 5. We find that in all the cases performance using activity regularization is worse compared with our best model shown in table 3.

Table 6: KL divergence of the Variational Bi-LSTM

| Dataset | PTB | Seq-MNIST | IMDB | TIMIT | Blizzard |
|---|---|---|---|---|---|
| KL | 0.001 | 0.02 | 0.18 | 3204.71 | 3799.79 |

## 2. Use of parametric encoder prior vs. fixed Gaussian prior

In our variational Bi-LSTM model, we propose to have the encoder prior over $\mathbf{z}_t$ as a function of the previous forward LSTM hidden state $\mathbf{h}_{t-1}$. This is done to omit the need of the backward LSTM during inference because it is unavailable in practical scenarios since predictions are made in the forward direction. However, to study whether the model learns to use this encoder or not, we record the KL divergence value of the best validation model for the various datasets. The results are reported in table 6. We can see that the KL divergence values are large in the case of IMDB, TIMIT and Blizzard datasets, but small in the case of Seq-MNIST and PTB. To further explore, we ran experiments on these datasets with fixed standard Gaussian prior like in the case of traditional VAE.

Interestingly we found that the model with fixed prior performed similarly in the case of PTB, but hurt performance in the other cases, which can be explained given their large KL divergence values in the original experiments.

## 5 RELATED WORK

Variational auto-encoders (Kingma & Welling, 2014) can be easily combined with many deep learning models. They have been applied in the feed-forward setting but they have also found usage in RNNs to better capture variation in sequential data (Sordoni et al., 2017; Fraccaro et al., 2016; Chung et al., 2015; Bayer & Osendorfer, 2014). VAEs consists of several muti-layer neural networks as probabilistic encoders and decoders and training is based on the gradient on log-likelihood lower bound (as the likelihood is in general intractable) of the model parameters $\Gamma$ along with a reparametrization trick. The derived variational lower-bound $\mathcal{L}_\Gamma$ for an observed random variable $\mathbf{x}$ is:

$$\log p(\mathbf{x}) \geq \mathcal{L}_\Gamma = \underset{q_\phi(\mathbf{z}|\mathbf{x})}{E}\left[\log \frac{p(\mathbf{x},\mathbf{z})}{q_\phi(\mathbf{z}|\mathbf{x})}\right] = \underset{q_\phi(\mathbf{z}|\mathbf{x})}{E}\left[\ln p_\theta(\mathbf{x}|\mathbf{z})\right] - D_{KL}(q_\phi(\mathbf{z}|\mathbf{x})\|p_\theta(\mathbf{z})), \quad (10)$$

where $D_{KL}$ denotes the Kullback-Leibler divergence and $p_\theta$ is the prior over a latent variable $\mathbf{z}$. The KL divergence tem can be expressed as the difference between the entropy of $q_\phi$ and the prior and fortunately, it can be computed and differentiated without estimation for some distribution families like Gaussians. Although maximizing the log-likelihood corresponds to minimizing the KL divergence, we have to ensure that the resulting $q_\phi$ remains far enough from an undesired equilibrium state where $q_\phi$ is almost everywhere equal to the prior over latent variables. Combining recurrent neural networks with variational auto encoders can lead to powerful generative models that are capable of capturing the variations in data, however, they suffer badly from this optimization issue as discussed by Bowman et al. (2015).

Recently, VAEs have also been applied to Bi-LSTMs by Sordoni et al. (2017) through a technique called Z-forcing. It is a powerful generative autoregressive model which is trained using the following variational evidence lower-bound

$$\mathcal{L}(\mathbf{x};\theta,\phi,\xi) = \sum_\ell \underset{q_\phi(\mathbf{z}_\ell|\mathbf{x})}{E}\left[\log p_\theta(\mathbf{x}_{t+1}|\mathbf{x}_{1:t},\mathbf{z}_{1:t})\right] - D_{KL}(q_\phi(\mathbf{z}_t|\mathbf{x})\|(p_\theta(\mathbf{z}_t|\mathbf{x}_{1:t-1},\mathbf{z}_{1:t-1}))$$

plus an auxiliary cost as a regularizer which is defined as $\log p_\xi(\mathbf{b}_t|\mathbf{z}_t)$. They show that the auxiliary cost helps in improving the final performance; however during inference the backward reconstructions have not been used in their approach. In our ablation study section below, we show experimentally that this connection is important towards improving the performance of Bi-LSTMs as is the case in our model.

## 6 CONCLUSION

Variational Bi-LSTMs are powerful autoregressive generative models that are capable of learning better representations by creating a channel to exchange the information of the past and the future. Moreover, the conditional distribution over the backward LSTM variables is learned that can lead to better learning results in practice. Furthermore, Variational Bi-LSTM model acts as a regularizer and makes both networks be informative enough to perform well on different benchmark problems taken from the literature.

## ACKNOWLEDGMENTS

The authors would like to thank Theano developers (Theano Development Team, 2016) for their great work.

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

APPENDIX

**A:** Derivation of variation lower bound $\mathcal{L}_{\boldsymbol{\Gamma}}$ in equation (2) in more details:

$$\log p(\mathbf{x}; \boldsymbol{\Gamma}) = \log \Big[ \prod_{t=0}^{T} \int_{\mathbf{z}_{1:T}} \int_{\tilde{\mathbf{b}}_t} \Big[ p_\eta(\mathbf{x}_{t+1}|\mathbf{x}_{1:t}, \mathbf{z}_{1:t}, \tilde{\mathbf{b}}_t) p_\psi(\tilde{\mathbf{b}}_t|\mathbf{z}_{1:t}) p_\theta(\mathbf{z}_t|\mathbf{x}_{1:t-1}, \mathbf{z}_{1:t-1}) \Big] d\tilde{\mathbf{b}}_t d\mathbf{z}_{1:T} \Big]$$

$$= \sum_{t=0}^{T} \log \Big[ \int_{\mathbf{z}_{1:T}} p_\theta(\mathbf{z}_t|\mathbf{x}_{1:t-1}, \mathbf{z}_{1:t-1}) \int_{\tilde{\mathbf{b}}_t} \Big[ p_\eta(\mathbf{x}_{t+1}|\mathbf{x}_{1:t}, \mathbf{z}_{1:t}, \tilde{\mathbf{b}}_t) p_\psi(\tilde{\mathbf{b}}_t|\mathbf{z}_{1:t}) \Big] d\tilde{\mathbf{b}}_t d\mathbf{z}_{1:T} \Big]$$

$$= \sum_{t=0}^{T} \log \Big[ \int_{\mathbf{z}_{1:T}} q_\phi(\mathbf{z}_t|\mathbf{x}) \frac{p_\theta(\mathbf{z}_t|\mathbf{x}_{1:t-1}, \mathbf{z}_{1:t-1})}{q_\phi(\mathbf{z}_t|\mathbf{x})} \int_{\tilde{\mathbf{b}}_t} \Big[ p_\eta(\mathbf{x}_{t+1}|\mathbf{x}_{1:t}, \mathbf{z}_{1:t}, \tilde{\mathbf{b}}_t) p_\psi(\tilde{\mathbf{b}}_t|\mathbf{z}_{1:t}) \Big] d\tilde{\mathbf{b}}_t d\mathbf{z}_{1:T} \Big]$$

$$\geq \sum_{t=0}^{T} \int_{\mathbf{z}_{1:T}} q_\phi(\mathbf{z}_t|\mathbf{x}) \log \Big[ \frac{p_\theta(\mathbf{z}_t|\mathbf{x}_{1:t-1}, \mathbf{z}_{1:t-1})}{q_\phi(\mathbf{z}_t|\mathbf{x})} \int_{\tilde{\mathbf{b}}_t} \Big[ p_\eta(\mathbf{x}_{t+1}|\mathbf{x}_{1:t}, \mathbf{z}_{1:t}, \tilde{\mathbf{b}}_t) p_\psi(\tilde{\mathbf{b}}_t|\mathbf{z}_{1:t}) \Big] d\tilde{\mathbf{b}}_t d\mathbf{z}_{1:T} \Big]$$

$$= \sum_{t=0}^{T} \int_{\mathbf{z}_{1:T}} \Big[ q_\phi(\mathbf{z}_t|\mathbf{x}) \log(\frac{p_\theta(\mathbf{z}_t|\mathbf{x}_{1:t-1}, \mathbf{z}_{1:t-1})}{q_\phi(\mathbf{z}_t|\mathbf{x})})$$

$$+ q_\phi(\mathbf{z}_t|\mathbf{x}) \log \Big[ \int_{\tilde{\mathbf{b}}_t} \Big[ p_\eta(\mathbf{x}_{t+1}|\mathbf{x}_{1:t}, \mathbf{z}_{1:t}, \tilde{\mathbf{b}}_t) p_\psi(\tilde{\mathbf{b}}_t|\mathbf{z}_{1:t}) \Big] d\tilde{\mathbf{b}}_t d\mathbf{z}_{1:T} \Big]$$

$$= \sum_{t=0}^{T} \Big[ \int_{\mathbf{z}_{1:T}} q_\phi(\mathbf{z}_t|\mathbf{x}) \log \Big[ \int_{\tilde{\mathbf{b}}_t} \Big[ p_\eta(\mathbf{x}_{t+1}|\mathbf{x}_{1:t}, \mathbf{z}_{1:t}, \tilde{\mathbf{b}}_t) p_\psi(\tilde{\mathbf{b}}_t|\mathbf{z}_{1:t}) \Big] d\tilde{\mathbf{b}}_t d\mathbf{z}_{1:T} \Big]$$

$$- D_{KL}(q_\phi(\mathbf{z}_t|\mathbf{x}) \| p_\theta(\mathbf{z}_t|\mathbf{x}_{1:t-1}, \mathbf{z}_{1:t-1})) \Big]$$

$$\geq \sum_{t=0}^{T} \Big[ \int_{\mathbf{z}_{1:T}} q_\phi(\mathbf{z}_t|\mathbf{x}) \int_{\tilde{\mathbf{b}}_t} p_\psi(\tilde{\mathbf{b}}_t|\mathbf{z}_{1:t}) \log \Big[ p_\eta(\mathbf{x}_{t+1}|\mathbf{x}_{1:t}, \mathbf{z}_{1:t}, \tilde{\mathbf{b}}_t) \Big] d\tilde{\mathbf{b}}_t d\mathbf{z}_{1:T}$$

$$- D_{KL}(q_\phi(\mathbf{z}_t|\mathbf{x}) \| p_\theta(\mathbf{z}_t|\mathbf{x}_{1:t-1}, \mathbf{z}_{1:t-1})) \Big]$$

$$\approx \sum_{t=0}^{T} \mathop{E}_{\mathbf{z}_{1:T} \sim q_\phi(\mathbf{z}|\mathbf{x})} \mathop{E}_{\tilde{\mathbf{b}}_t \sim p_\psi(\tilde{\mathbf{b}}_t|\mathbf{z}_t)} \Big[ \log p_\eta(\mathbf{x}_{t+1}|\mathbf{x}_{1:t}, \mathbf{z}_{1:t}, \tilde{\mathbf{b}}_t) \Big] - D_{KL}(q_\phi(\mathbf{z}_t|\mathbf{x}) \| p_\theta(\mathbf{z}_t|\mathbf{x}_{1:t-1}, \mathbf{z}_{1:t-1})),$$

