# OpenReview forum: "Variational Bi-LSTMs"
_ICLR.cc/2018/Conference — Reject_

### Official Review · AnonReviewer3 · 2017-11-25
**Interesting ideas; paper could be improved by more ablation experiments, theoretical justifications, and evaluation methods**

**Rating:** 4
**Confidence:** 4

**Review:**

*Quality*

The paper is easy to parse, with clear diagrams and derivations at the start. The problem context is clearly stated, as is the proposed model.

The improvements in terms of average log-likelihood are clear. The model does improve over state-of-the-art in some cases, but not all.

Based on the presented findings, it is difficult to determine the quality of the learned models overall, since they are only evaluated in terms of average log likelihood. It is also difficult to determine whether the improvements are due to the model change, or some difference in how the models themselves were trained (particularly in the case of Z-Forcing, a closely related technique). I would like to see more exploration of this point, as the section titled “ablation studies” is short and does not sufficiently address the issue of what component of the model is contributing to the observed improvements in average log-likelihood.

Hence, I have assigned a score of "4" for the following reasons: the quality of the generated models is unclear; the paper does not clearly distinguish itself from the closely-related Z-Forcing concept (published at NIPS 2017); and the reasons for the improvements shown in average log-likelihood are not explored sufficiently, that is, the ablation studies don't eliminate key parts of the model that could provide this information.

More information on this decision is given in the remainder.

*Clarity*

A lack of generated samples in the Experimental Results section makes it difficult to evaluate the performance of the models; log-likelihood alone can be an inadequate measure of performance without some care in how it is calculated and interpreted (refer, e.g., to Theis et al. 2016, “A Note on the Evaluation of Generative Models”).

There are some typos and organizational issues. For example, VAEs are reintroduced in the Related Works section, only to provide an explanation for an unrelated optimization challenge with the use of RNNs as encoders and decoders.

I also find the motivations for the proposed model itself a little unclear. It seems unnatural to introduce a side-channel-cum-regularizer between a sequence moving forward in time and the same sequence moving backwards, through a variational distribution. In the introduction, improved regularization for LSTM models is cited as a primary motivation for introducing and learning two approximate distributions for latent variables between the forward and backward paths of a bi-LSTM. Is there a serious need for new regularization in such models? The need for this particular regularization choice is not particularly clear based on this explanation, nor are the improvements state-of-the-art in all cases. This weakens a possible theoretical contribution of the paper.

*Originality*

The proposed modification appears to amount to a regularizer for bi-LSTMs which bears close similarity to Z-Forcing (cited in the paper). I recommend a more careful comparison between the two methods. Without such a comparison, they are a little hard to distinguish, and the originality of this paper is hard to evaluate. Both appear to employ the same core idea of regularizing an LSTM using a learned variational distributions. The differences *seem* to be in the small details, and these details appear to provide better performance in terms of average log-likelihood on all tasks compared to Z-Forcing--but, crucially, not compared to other models in all cases.

---

> ### Author Response · Authors · 2017-12-21
> **Clarifications and additional experimental results**
>
> Thank you for your detailed comments. The major concern of the reviewer seems to be the lack of a clear contrast between our proposed model and Z-Forcing, which is closely related to our model. We will try to make this distinction clearer. In order to showcase the differences between Z-Forcing and our model in terms of the differences between how our model is trained, rather than just the architectural differences (as stressed by the reviewer in their comments), we additionally conducted the following experiment which incrementally adds the additional optimization changes to Z-Forcing (that we added to train our model). Specifically we run experiments to see the effects of stochastic backprop on Z-Forcing. We also add a reconstruction cost on h_t in the Z-Forcing model as another separate experiment. So for a detailed comparison, we show the evolution of Bits Per Character (BPC) on PTB for four cases:
> 1. Z-forcing
> 2. Z-forcing + stochastic backprop (on the auxiliary cost)
> 3. Z-forcing + stochastic backprop (on the auxiliary cost) + reconstruction/auxiliary loss
> 4. Variational Bi-LSTM
> The plot can be found in this anonymous link https://anonfile.com/HdFdcadbb6/ptb_sdc_zf_rec.png . As can be seen there is a gradual improvement from model 1 to model 4.
> Further, we also have the following ablation studies in the latest version (https://anonfile.com/W6i9bad3b4/ICLR18_VLM.pdf) of our paper that the
>  Reconstruction loss on h_t  vs activity regularization on h_t-- here we show how the auxiliary reconstruction loss on h_t performs between compared with simply using an l2 regularization on h_t.
>  Use of parametric encoder prior vs. fixed (standard VAE) Gaussian prior-- here we discuss the importance of the VAE prior we propose (which is conditional over h_t) compared to a fixed Gaussian that is usually used in VAEs.
>  Effectiveness of auxiliary costs and stochastic back-propagation-- here we show that stochastic backpropagation helps during optimization.
>  Importance of sampling from VAE prior during training-- here we show that sampling z_t during training has a regularization effect on the model.
>
> To address the reviewer’s concern regarding additional qualitative analysis of the data generated by our mode, here are some of the samples generated by our model on the IMDB dataset:
> it was very well directed by the critics and critics who have n't seen it .
> i did n't want to see this movie .
> but the movie does have a few laughs .
> the action is also very well acted but it has a great story .
> it 's just a bit too slow and the ending is very good .
> this film is not as bad as you 've heard .
> it 's also quite a good film with a great cast and great story lines .
> and what the movie was nominated for is a great cast .
> it 's a good film and you ca n't miss it .
>
> Regarding your concern for the need of a new regularizer, our reasoning is as follows. In the paper we already mention that the forward and backward LSTM capture different aspects of a temporal sequence, and this is the reason why (for traditional Bi-LSTMs) concatenating the hidden representations from the two LSTMs leads to better performance in tasks where such a concatenation is possible. However, this concatenation is not possible in the next step prediction tasks (Eg. language generation) where only the forward LSTM must be used during inference. Hence the information captured by the backward LSTM in a Bi-LSTM trained separate from the forward LSTM is lost. For this reason, an objective/regularization that jointly optimizes the two LSTMs in a Bi-LSTM is needed. Other examples of such joint optimization are Z-Forcing and twin networks that we cite in our paper.

---

### Official Review · AnonReviewer1 · 2017-11-27
**An interesting paper, but not the clearest presentation.**

**Rating:** 7
**Confidence:** 3

**Review:**

This paper proposes a particular form of variational RNN that uses a forward likelihood and a backwards posterior.  Additional regularization terms are also added to encourage the model to encode longer term dependencies in its latent distributions.

My first concern with this paper is that the derivation in Eq. 1 does not seem to be correct.  There is a p(z_1:T) term that should appear in the integrand.

It is not clear to me why h_t should depend on \tilde{b}_t.  All paths from input to output through \tilde{b}_t also pass through z_t so I don't see how this could be adding information.  It may add capacity to the decoder in the form of extra weights, but the same could be achieved by making z_t larger. Why not treat \tilde{b}_t symmetrically to \tilde{h}_t, and use it only as a regularizer?

In the no reconstruction loss experiments do you still sample \tilde{b}_t in the generative part?  Baselines where the \tilde{b}_t -> h_t edge is removed would be very nice.

It seems the Blizzard results in Figure 2 are missing no reconstruction loss + full backprop.

I don't understand the description of the "Skip Gradient" trick.  Exactly which gradients are you skipping at random?

Do you have any intuition for why it is sometimes necessary to set beta=0?

---

> ### Author Response · Authors · 2017-12-21
> **Clarifications**
>
> Thank you for your constructive comments. You are right. We have corrected Eq 1 in our latest version (https://anonfile.com/W6i9bad3b4/ICLR18_VLM.pdf). Please note that our model implementation was not affected by these writing mistakes.
>
> To answer why it is beneficial to make h_t dependent on \tilde{b}_t, note that forward and backward LSTMs model data sequences independently in different ways in a traditional Bi-LSTM. Creating the dependence from \tilde{b}_t to h_t is important to make the forward LSTM use of information from the backward LSTM and thus learn a richer representation. This representation is useful in tasks like next step prediction where only the forward LSTM is used during inference and hence the structure captured by the backward LSTM is lost in the case of a traditional Bi-LSTM. In our model this structure is utilized.
>
> We did experiments where we remove the connection from \tilde{b}_t to h_t and found that only using \tilde{b}_t in the reconstruction cost (as a regularizer) does not produce as good results as our model where both the reconstruction and feeding \tilde{b}_t to h_t is used. Thus feeding \tilde{b}_t to h_t helps the forward model during inference. On the flip side, we do not pass \tilde{h}_t to b_t because we do not use the backward LSTM during inference, and so it may not benefit us.
>
> Yes, in the no reconstruction loss experiments we do sample \tilde{b}_t.
>
> We have uploaded the Blizzard results in Figure 2 with no reconstruction loss + full backprop that you asked for to this anonymous link https://anonfile.com/j3nbo0dbbd/blz_rec_sdc_full.png It can be seen that reconstruction loss with stochastic backprop yields the best performance compared to all other alternatives.
>
> Regarding setting \beta=0, we treat it as a hyperparameter and so it is chosen using the validation set. We do not have any explanation why having it zero is better sometimes.
>
> We have made the description of skip gradient clearer in the latest version. The idea is to stochastically skip gradients of the auxiliary reconstruction costs with respect to the recurrent
> units from back-propagating through time. To achieve this, at each time step, a mask drawn from a Bernoulli distribution which governs whether to skip the gradient or to back-propagate it for each data sample.

---

### Official Review · AnonReviewer2 · 2017-11-27
**Propose explicitly modeling the hidden state of the backward RNN used for inference in a sequential deep generative model. Good idea, good empirical performance; the writing, however, is confusing.**

**Rating:** 6
**Confidence:** 4

**Review:**

This paper builds a sequential deep generative model with (1) an inference network parameterized by an RNN running from the future to the past and (2) an explicit representation of the hidden state of the backward RNN in the generative model. The model is validated on held-out likelihood via the ELBO on text, handwriting, speech and images. It presents good emprical results and works at par with or better than many other baselines considered.

The main source of novelty here the choice made in the transition function of z_t to also incorporate an explicit variable to models the hidden state of the backward RNN at inference time and use that random variable in the generative process. This is a choice of structural prior for the transition function of the generative model that I think lends it more expressivity realizing the empirical gains obtained.

I found the presentation of both the model and learning objective to be confusing and had a hard time following it. The source of my confusion is is that \tilde{b} (the following argument applies equivalently to \tilde{h}) is argued to be a latent variable. Yet it is not inferred (via a variational distribution) during training.

Please correct me if I'm wrong but I believe that an easier to understand way to explain the model is as follows: both \tilde{b} and \tilde{h} should be presented as *observed* random variables during *training* and latent at inference time. Training then comprises maximizing the marginal likelihood of the data *and* maximizing the conditional likelihood of the two observed variables(via p_psi and p_eta; conditioned on z_t). Under this view, setting beta to 0 simply corresponds to not observing \tilde{h_t}. alpha can be annealed but should never be set to anything less than 1 without breaking the semantics of the learned generative model.

Consider Figure 1(b). It seems that the core difference between this work and [Chung et. al] is that this work parameterizes q(Z_t) using x_t....x_T (via a backward RNN). This choice of inference network can be motivated from the point of view of building a better approximation to the structure of the posterior distribution of Z_t under the generative model. Both [Fracarro et. al] and [Krishnan et. al] (https://arxiv.org/pdf/1609.09869.pdf) use RNNs from x_T to x_1 to train sequential state space models. [Gao et. al] (https://arxiv.org/pdf/1605.08454.pdf) derive an inference network with a block-diagonal structure motivated by correlations in the posterior distribution. Incorporating a discussion around this idea would provide useful context for where this work stands amongst the many sequential deep generative models in the
literature.

Questions for the authors:
* How important is modeling \tilde{h_t} in TIMIT, Blizzard and IMDB?
* Did you try annealing the KL divergence in the PTB experiment. Based on the KL divergence you report it seems the latent variable is not necessary.

Overall, I find the model to be interesting and it performs well empirically. However, the text of the paper lacks a bit of context and clarity that makes understanding it challenging to understand in its current form.

---

> ### Author Response · Authors · 2017-12-21
> **Comments and explanations**
>
> Thank you for your positive comments. Indeed, making the forward LSTM ‘aware’ of the backward LSTM’s state is a crucial factor in improving the expressivity of our model.
>
> We apologize for the lack of clarity in the submitted version; we have made the text clearer in our latest version. To clarify the doubt you mentioned about \tilde{b}, we do sample \tilde{b}_t during training from p_{\psi}(\tilde{b}_t | z_t), and feed it to h_t, where z_t is sampled from q_\phi (z_t | h_{t-1}, b_t).  This process of inferring \tilde{b}_t and feeding to h_t however is implicitly captured in the term p(x_{t+1} | h_t ). The only \tilde{b}_t dependent term that appears in the objective is p_{\psi}(b_t | z_t ), which (to be precise) maximizes p_{\psi}( \tilde{b}_t = b_t | z_t ).
>
> Regarding your comment “both \tilde{b} and \tilde{h} should be presented as *observed* random variables during *training* and latent at inference time”, that’s true.
>
> Regarding your comment about not setting alpha to anything less than 1, we are not sure if we understand your concern correctly. We treat alpha as a hyperparameter, so its value should be chosen based on validation set. The positive value of alpha governs how much weightage is given to the reconstruction of b_t vs the rest of the terms in the cost.
>
> As suggested by the reviewer, here is a brief comparison between our model and the papers cited by the reviewer:
> In krishnan et al, the data x_t at each time step t is modeled using a VAE with hidden state z, where the approximate posterior q_{\phi} (z | x) is a function of the forward and backward hidden states, and the KL divergence minimizes the difference between this approximate posterior and the prior over z. The key difference between their model and ours is that their model learns a VAE on the data space, i.e., the reconstruction error is on the data itself, such that the latent variable z of the VAE is a function of the Bi-RNN's hidden states. In our model on the other hand, the VAE is learned on the Bi-LSTM's hidden state, i.e., the reconstruction error is on the forward and backward LSTM's hidden states h_t and b_t which share the latent variable z_t. In Gao et al, the approximate prior at time step t is modeled as q_{\phi} (z_t | z_{t-1}, x_t ), which factorizes as q_{\phi} (z_t | z_{t-1} ) . q_{\phi} (z_t | x_t ). Each of the latter two functions are modeled as Gaussians with mean and variance as a non-linear function of z_{t-1} and x_t respectively. Thus this model does not make use of recurrent neural networks in modeling the data. Secondly, similar to Krishnan et al, this model learns to reconstruct data instead of a hidden space, as in our model.
>
> We did try experiments without modeling \tilde{h}_t but found the results to be slightly worse. We believe it acts as a regularizer on the activation h_t learned by the model. But in general the coefficient \beta used for the reconstruction loss of h_t is a hyperparameter and so it should be chosen using the validation set.
> Indeed, in the ablation studies, we report that the KL term is not useful in the case of PTB dataset because the KL term is small and performance remains unaffected when not including it in the objective. But performance drops in the case of the other datasets if the KL term is removed since for these datasets the KL term is large.
>
> Once again, we apologize for the lack of clarity. We have made the text clearer in the latest version of our paper which can be found at the anonymous link (https://anonfile.com/W6i9bad3b4/ICLR18_VLM.pdf).

---

> > ### Comment · AnonReviewer2 · 2018-01-08
> > **Updated Review**
> >
> >
> > After reading the author's response, R3's review, and the revised paper, I am more concerned about the clarity in this work than I was initially.
> >
> > "Regarding your comment .... that’s true...... We treat alpha as a hyperparameter, so its value should be chosen based on validation set. The positive value of alpha governs how much weightage is given to the reconstruction of b_t vs the rest of the terms in the cost."
> >
> > One my concerns that still remains from my initial review is that there is no probabilistic reason provided for using alpha as a hyperparameter.
> >
> > Let me try and illustrate why I think alpha (and beta, but for the moment, we'll ignore the latent variable \tilde{h_t-1} and focus just on \tilde{b}_t) should have the value 1.
> > Consider a simplification of the model that looks at a single time slice of the model -- i.e. just the variables z_t, \tilde{b}_t and h_t in Figure 1(b) (we'll refer to them as z, b, and h respectively).
> >
> > For a single time-slice variant of this model, the joint distribution is p(z)p(b|z)p(h|b,z).
> > As mentioned earlier and acknowledged, at training time, b and h are observed while z is latent. So if we want to maximize the likelihood of the observed data during training, we have:
> > \log p(b,h) = \log \int_z p(b,h,z) = \log \int_z  p(b|z) p(h|b,z) p(z) * q(z|h)/q(z|h) \geq (Jensens) E_{q(z|h)}[\log p(b|z)+ \log p(h|b,z)] - KL[q(z|h)||p(z)]
> >
> > Contrast this with Equation (10) in the paper. Note that using alpha <1 implies that we multiply a negative number (namely log p(b|z)) with a fraction which always *increases* the number artificially. This means the resulting objective is no longer a valid lower-bound on the marginal likelihood of the observed data.
> > While I can potentially see a case for annealing alpha to 1, I'm a little concerned by the numbers reported when alpha is set to 0.0001.
> > What is the number reported at test time -- dose it use alpha? Please do correct me if I've missed something and there is a probabilistic reason for why alpha can be <1; as far as I can tell, it has only appeared in Equation (10) and not before in Equation (5).
> >
> > Overall, while I think the paper outlines an interesting idea,
> > in its current form (in the revised version) I still find it difficult to follow and not appropriately motivated
> > or set in context of recent work (see also comments by R3). Finally, a minor point -- At the bottom of Page 4 there is a paragraph about a heuristic used at training time.
> > Please expand on this further if you found it useful by explicitly stating how it changes the lower-bound at training time.
> > Please also use a different name than "Stochastic Backpropagation" which has been used before in https://arxiv.org/abs/1401.4082.

---

> > > ### Author Response · Authors · 2018-01-16
> > > **Clarification of the proposed objective**
> > >
> > > We apologize for the confusion. If we consider the Jensen's inequality derived from the term \log p(b,h) as a starting point, then your argument about using alpha and beta equal to 1 would be absolutely correct. However, we do not have the term \log p(b,h) in the original objective (equation 4). To arrive at our objective, consider that we have designed the architecture as shown in figure 1. Then we derive equation 10 from equation 5 as follows:
> > >
> > > 1. The term  p(x_{t+1} | x_{1:t}, z_t, \tilde{b}_t) in equation 5 is instantiated with p(x_{t+1} | h_t) in equation 10.
> > > 2. The KLD term in equation 10 is an instantiation of the KLD term in equation 5 based on our architecture.
> > > 3. Finally, we add 3 regularization terms: p(x_{t+1} | b_t) + alpha p(b_t | z_t) + beta p(h_t | z_t)
> > >
> > > Notice that using only the first two steps as our objective leads to a b_t that is a deterministic but random function of x_t and b_{t-1} depending completely on the initialization of the weights of the backward LSTM. The 3rd step adds two regularizations related to b_t and a regularization on h_t. In other words, you are right in pointing out that equation 10 is not exactly the lower bound stated in equation 5, but rather it is one with added regularization terms which help the model generalize better. Specifically, the regularization term p(x_{t+1}|b_t) helps b_t learn information about future, the second regularization term alpha p(b_t | z_t) makes \tild{b_t} learn to be close to b_t that can help predict x_{t+1}, and the last regularization term beta p(h_t | z_t) regularizes h_t by imposing a reconstruction loss.
> > >
> > > Thank you for pointing it out that the name “stochastic gradient” can be misleading since it has been used previously. We have changed it to “skipping gradient” in the latest version of our paper.

---

### Public Comment · (anonymous) · 2017-11-20
**Difference to Z-Forcing ?**

Hello Authors,

Very interesting work!

I have been trying to reproduce your experiments. As far as I understand it is straightforward extension to Z-Forcing(https://arxiv.org/abs/1711.05411). I tried to replicate your results using the Z-Forcing code(https://github.com/sordonia/zforcing) so far I have not been able to replicate your results. Adding the reconstruction cost (in the forward RNN, which was also missing from Z-Forcing) does not seem to have any impact on results.

So are you doing something which is not mentioned in the paper?

---

> ### Author Response · Authors · 2017-11-21
> **Probable causes of your issue**
>
> We appreciate your interest in our paper and your effort to reproduce our results.
> We apologize for the lack of clarity in the submitted version. We have improved the model description in our current version. We would like to point out that while our model is similar in spirit to Z-forcing, there are notable differences in the derivation of the variational lower bound and the auxiliary costs that provide improvement in performance because the forward LSTM is implicitly directly fed the backward LSTM's state which is in contrast with Z-forcing.
>
> From your comment, it seems to us that you add a reconstruction cost on h_t on top of the Z-forcing objective. If this is true, then we would like to clarify that in addition to adding the reconstruction cost and feeding z_t to h_t, we also pass \tilde{b}_t to h_t. Amongst other differences, this is a crucial difference between Z-forcing and our model. In other words, during training, we sample \tilde{b}_t for a sampled z_t, and encourage this \tilde{b}_t to be similar to b_t, and also feed this \tilde{b}_t to h_t. In this way, our model learns to implicitly use b_t during training as an input to h_t. This is different from Z-forcing where the model passes z_t to h_t while minimizing the KL divergence difference between the prior and posterior over z_t.
>
> Another possible difference in your implementation could be that we suggest using stochastic backpropagation through the auxiliary costs. This entails that the gradients through the auxiliary cost be stochastically dropped during training.
>
> We hope these suggestions help in reproducing the results we report in our paper.
>
> For further clarification, we have uploaded an anonymous copy of the latest version of our paper here: https://anonfile.com/W6i9bad3b4/ICLR18_VLM.pdf.

---

> > ### Public Comment · (anonymous) · 2017-11-21
> > **Thanks for clarification!**
> >
> > "Another possible difference in your implementation could be that we suggest using stochastic back-propagation through the auxiliary costs. This entails that the gradients through the auxiliary cost be stochastically dropped during training"
> >
> > I was not dropping these gradients, this is the only difference I could think of. Though, this raises the point, is all the benefit actually coming from this "stochastic back-propagation" over Z-Forcing ? As without using this stochastic back-propagation, results seems more or less same to Z-Forcing (https://arxiv.org/abs/1711.05411)
> >
> > I'd encourage the authors to add the results with/without "stochastic back-propagation"  and compare themselves to the results which Z-Forcing paper reports.
> >
> > Another thing which would make this submission strong,  is to analyze how useful the latents (learned z's) are. For ex. may be for some classification task.

---

> > > ### Author Response · Authors · 2017-11-23
> > > **Further experiments for clarification**
> > >
> > > The analysis of our method with and without stochastic backprop as well as with and without reconstruction losses are provided in the ablation studies section (figure 2). The text around this figure is unfortunately missing in the submitted version but can be found in the latest (anonymous) version we have linked in our previous reply. This analysis shows how our model benefits from stochastic backprop.
> > >
> > > We had also run experiments to see the effects of stochastic backprop on Z-forcing. We additionally also add a reconstruction cost on h_t in the Z-forcing model as another separate experiment. So for a detailed comparison, we show the evolution of BPC on PTB for four models:
> > > 1. Z-forcing
> > > 2. Z-forcing + stochastic backprop (on the auxiliary cost)
> > > 3. Z-forcing + stochastic backprop (on the auxiliary cost) + reconstruction/auxiliary loss
> > > 4. Variational Bi-LSTM
> > > The plot can be found in this anonymous link https://anonfile.com/HdFdcadbb6/ptb_sdc_zf_rec.png . As can be seen there is a gradual improvement from model 1 to model 4.
> > >
> > > We agree with your suggestion of exploring the usefulness of the latent variable z and we ourselves had given thought to it. However, this is not the focus of our work, and this analysis applies to all models that make use of a latent variable in LSTMs (including Z-forcing). So we leave this as separate future work.

---

### Public Comment · (anonymous) · 2017-11-22
**Some questions about Eq. 10 in latest version**

Dear authors,

it's an interesting research, but I still have some questions about the objective, and I hope you can give me a help!

In Eq.10, the objective, \tilde{b}_t is sampled from p_psi. However, there is no \tilde{b}_t in the inside term, so it seems that there is no need to sample \tilde{b}_t? Is it just a typo?

Next, based on figure 1(a) and your answer, I think there may be some terms to stand for the direct connection between h_t and \tilde{b}_t. However, it seems that, in Eq. 10 , there is no term to stand for the directly conditional dependence between h_t and \tilde{b}_t( or b_t ). I guess maybe the term p(x_{t+1} | b_t) includes relations like p(x_{t+1} | h_t)p(h_t | b_t), is it true?
Thanks for your help!

---

> ### Author Response · Authors · 2017-11-23
> **Clarification for Eq. 10**
>
> We thank you for raising this question. Your first doubt regarding \tilde{b}_t is because of ambiguity in our notation. We do sample \tilde{b}_t from p_psi. These samples are used in the third term-- alpha log p_psi(b_t | z_t). To remove ambiguity, this term should be read as-- alpha log p_psi(\tilde{b}_t = b_t | z_t).
>
> Regarding your second question about terms that should relate \tilde{b}_t and h_t, we believe the notations are correct. Imagine if we were to write the objective for a simple LSTM, then this objective would simply contain a summation of terms p(x_{t+1} | h_t) over time steps t. The dependence of h_t on the previous time steps are implicit. Similarly, in our objective, the term p(x_{t+1} | h_t) implicitly contains the dependence on \tilde{b}_t, z_t and the previous time step variables.

---

### Decision · Program_Chairs · 2018-01-29
**ICLR 2018 Conference Acceptance Decision**

**Decision:**

Reject

**Comment:**

This paper proposes a method for performing stochastic variational inference for bidirectional LSTMs through introducing an additional latent variable that induces a dependence between the forward and backward directions.  The authors demonstrate that their method achieves very strong empirical performance (log-likelihood on test data) on the benchmark TIMIT and BLIZZARD datasets.

The paper is borderline in terms of scores with a 7, 6 and 4.  Unfortunately the highest rating also corresponds to the least thorough review and that review seems to indicate that the reviewer found the technical exposition confusing.  AnonReviewer2 also found the writing confusing and discovered mistakes in the technical aspects of the paper (e.g. in Eq 1).  Unfortunately, the reviewer who seemed to find the paper most easy to understand also gave the lowest score.  A trend among the reviewers and anonymous comments was that the paper didn't do a good enough job of placing itself in the context of related work (Goyal et. al, "Z-forcing") in particular.  The authors seem to have addressed this (curiously in an anonymous link and not in an updated manuscript) but the manuscript itself has not been updated.

In general, this paper presents an interesting idea with strong empirical results.   The paper itself is not well composed, however, and can be improved upon significantly.  Taking the reviews into account and including a better treatment of related work in writing and empirically will make this a much stronger paper.

Pros:
- Strong empirical performance (log-likelihood on test data)
- A neat idea
- Deep generative models are of great interest to the community

Cons:
- Incremental in relation to Goyal et al., 2017
- Needs better treatment of related work
- The writing is confusing and the technical exposition is not clear enough